# Performance of Different Accelerometry-Based Metrics to Estimate Oxygen Consumption during Track and Treadmill Locomotion over a Wide Intensity Range

**DOI:** 10.3390/s23115073

**Published:** 2023-05-25

**Authors:** Henri Vähä-Ypyä, Jakob Bretterhofer, Pauliina Husu, Jana Windhaber, Tommi Vasankari, Sylvia Titze, Harri Sievänen

**Affiliations:** 1The UKK Institute for Health Promotion Research, 33500 Tampere, Finland; henri.vaha-ypya@ukkinstituutti.fi (H.V.-Y.); pauliina.husu@ukkinstituutti.fi (P.H.); tommi.vasankari@ukkinstituutti.fi (T.V.); 2Institute of Human Movement Science, Sport and Health, University of Graz, 8010 Graz, Austriasylvia.titze@uni-graz.at (S.T.); 3Sports Medicine Performance and Movement Analysis, Medical University of Graz, 8010 Graz, Austria; jana.windhaber@medunigraz.at

**Keywords:** accelerometer, adults, energy consumption, estimation, intensity, measurement, physical activity, running, walking

## Abstract

Accelerometer data can be used to estimate incident oxygen consumption (VO_2_) during physical activity. Relationships between the accelerometer metrics and VO_2_ are typically determined using specific walking or running protocols on a track or treadmill. In this study, we compared the predictive performance of three different metrics based on the mean amplitude deviation (MAD) of the raw three-dimensional acceleration signal during maximal tests performed on a track or treadmill. A total of 53 healthy adult volunteers participated in the study, 29 performed the track test and 24 the treadmill test. During the tests, the data were collected using hip-worn triaxial accelerometers and metabolic gas analyzers. Data from both tests were pooled for primary statistical analysis. For typical walking speeds at VO_2_ less than 25 mL/kg/min, accelerometer metrics accounted for 71–86% of the variation in VO_2_. For typical running speeds starting from VO_2_ of 25 mL/kg/min up to over 60 mL/kg/min, 32–69% of the variation in VO_2_ could be explained, while the test type had an independent effect on the results, except for the conventional MAD metrics. The MAD metric is the best predictor of VO_2_ during walking, but the poorest during running. Depending on the intensity of locomotion, the choice of proper accelerometer metrics and test type may affect the validity of the prediction of incident VO_2_.

## 1. Introduction

Accelerometer-based assessment of physical activity (PA) intensity is widely used in observational and intervention studies. Accelerometers permit continuous assessment of PA intensity patterns over long periods with high resolution. A crucial step in the assessment of PA is the extraction of the useful information from the raw acceleration signal and transforming it into meaningful measures, such as oxygen consumption (VO_2_) or metabolic equivalent (MET). While the acceleration signal obtained from the sensor follows the biomechanical rules determined by body dimensions, attachment position, and movement efficiency, the corresponding energy cost is affected by physiological maturation and efficiency of the given person [1].

Current ways to monitor daily physical behavior with accelerometer-based methods vary a lot in terms of the sensor itself, its wear time and placement, and analysis methods used to extract the information of interest. This methodological variance is likely the factor that feeds the current controversies in the epidemiology of daily physical behavior. Chastin et al. concluded recently that the data from hip-worn accelerometry can provide accurate and meaningful estimates of PA [2]. Further, recent advances in data analysis have made it possible to quantify also the sedentary time spent in standing, sitting, and reclining postures [3,4].

Relationships between the accelerometer output, gait speed, and VO_2_ are typically determined using specific walking or running protocols on a treadmill or overground. So far, only a few studies have compared these two approaches [5,6]. Yngve et al. found that during track locomotion the accelerometer output was consistently higher and VO_2_ slightly lower compared to treadmill locomotion when both activities were performed at the same individual velocity [5]. Barnett et al., in turn, found that the treadmill-based calibration equations overestimated both VO_2_ and walking speed during a free-living test, whereas the calibration equations based on acceleration data collected on a 400 m track at a controlled speed provided accurate and unbiased estimates of VO_2_ [6]. The above observations mean that the accelerometer outputs differ whether the locomotion was performed on the track or treadmill. Thus, the differences in gait mechanics between treadmill walking and overground walking appear to result in inconsistent estimations of free-living gait speed and VO_2_ using the accelerometer data [5,6].

We have devised the mean amplitude deviation (MAD) of the resultant acceleration signal as a metric for comparable classification of PA intensity irrespective of substantial differences in measurement ranges and sampling rates of different accelerometer brands [7]. Several other researchers have evaluated the performance of the MAD metric and found it at least satisfactory [8,9,10,11,12,13,14,15]. The initial MAD method provides a valid and accurate estimate of incident VO_2_ within a wide range of walking and running speeds on track locomotion [16]. However, Chen et al. recently stated that the MAD method employing trunk-worn accelerometer data cannot be used to estimate VO_2_ while running on a treadmill [14]. In this study, we therefore assessed the validity of three different MAD-based analysis methods employing hip-worn accelerometer data in estimating VO_2_ during track and treadmill locomotion.

## 2. Materials and Methods

### 2.1. Accelerometry

This study employed the raw acceleration signals from a triaxial accelerometer (Hookie AM20, Traxmeet Ltd., Espoo, Finland) collected during controlled track and treadmill tests. This accelerometer employs the commonly used 13-bit digital triaxial acceleration sensor (ADXL345; Analog Devices Inc, Norwood, MA, USA). The measurement range of the accelerometer was ±16 *g* (*g* denotes for Earth’s gravity, 9.81 m/s^2^) and the data were measured at a 100 Hz sampling frequency.

The accelerometer was attached to the hip-mounted elastic belt at the level of the iliac crest. Because the indoor track had banked turns only to left, the track group kept the accelerometer either at the right or left side of the hip as per random assignment. The left side (i.e., the inner side of the curve) was assigned to 13 participants and the right side (i.e., the outer side of the curve) to 16 participants. The treadmill group kept the accelerometer on the right side of the hip.

### 2.2. Test Protocols

The track test was conducted by the UKK Institute and the treadmill test by the University of Graz. The track group consisted of 29 healthy volunteers (15 males and 14 females), and the treadmill group of 24 healthy volunteers (12 males and 12 females). Before testing, participants’ body height and weight were measured with standard methods. Participants were informed about the experimental test protocol, and they gave their written informed consent before the tests. Local ethical committees approved the studies.

The track group performed a pace-conducted non-stop test on a 200 m long indoor track. The initial speed of the track test was 0.6 m/s (2.16 km/h), and the speed was increased by 0.4 m/s (1.44 km/h) every 2.5 min.

The initial speed of the treadmill test was 2.0 km/h, and the speed was increased by 2.0 km/h every 3.0 min. The treadmill grade was constantly 1.5%. At the end of each stage, the treadmill test was paused for 30 s for blood sampling.

While performing the test, the participants could freely decide whether they preferred walking or running for the given speed of locomotion. The test was continued until volitional exhaustion when the participant could not keep up with the concurrent pace.

During the track test, VO_2_ was continuously measured in breath-by-breath mode using a portable metabolic gas analyzer (Oxycon Mobile, Carefusion, Yorba Linda, CA, USA), and the data were recorded with a telemetry system. During the treadmill test, VO_2_ was continuously measured in breath-by-breath mode using a cardiopulmonary exercise testing system (Oxygon Pro^®^, Carl Reiner GmbH, Vienna, Austria). Both devices were calibrated before each test according to the manufacturer’s instructions.

### 2.3. Data Analysis

The acceleration signal was analyzed for the final two minutes of each stage, when the participants had reached a steady rhythm. As the steady-state value of VO_2_ for the given speed, the mean VO_2_ of the final minute of the corresponding stage was used. Stages with a respiratory exchange ratio over 1.0 or not fully completed were excluded from the analysis.

According to our standard procedure using the MAD method [7,16,17,18], the accelerometer data were analyzed in six-second epochs. For each epoch, MAD, MADxyz, and mean magnitude (MM) values were calculated using Equations (1)–(4). In these equations, *r_i_* is the magnitude of the incident resultant acceleration of the three orthogonal vectors *x_i_*, *y_i_*, and *z_i_*; *N* is the number of samples in the epoch (for the six-second epoch 600); and *R_ave_*, *X_ave_*, *Y_ave_*, and *Z_ave_* are the mean acceleration values of the epoch. The unit of all calculated values is milligravity (m*g*), which corresponds to one-thousandth of Earth’s gravitational force.
(1)$$r_{i}=\sqrt{{x_{i}}^{2}+{y_{i}}^{2}+{z_{i}}^{2}}$$
(2)$$MAD=\frac{1}{N}\sum_{i=1}^{N}\left|r_{i}-R_{ave}\right|$$
(3)$$MAD_{xyz}=\sqrt{{\left(\frac{1}{N}\sum_{i=1}^{N}\left|x_{i}-X_{\mathrm{ave}\;}\right|\right)}^{2}+{\left(\frac{1}{N}\sum_{i=1}^{N}\left|y_{i}-Y_{\mathrm{ave}\;}\right|\right)}^{2}+{\left(\frac{1}{N}\sum_{i=1}^{N}\left|z_{i}-Z_{\mathrm{ave}\;}\right|\right)}^{2}}$$
(4)$$MM=R_{ave}-\sqrt{{X_{ave}}^{2}+{Y_{ave}}^{2}+{Z_{ave}}^{2}}$$

For illustration, Figure 1 depicts the measured raw and processed acceleration data for a two-second walking period. The conventional MAD metric is sensitive to the variation in the total magnitude of the acceleration, while the MADxyz metric and the novel MM metric are also sensitive to changes in the accelerometer orientation relative to the Earth’s gravity vector. The MM metric utilizes the same acceleration parameters which are needed to calculate not only the MAD and MADxyz values used in this study but also to determine the angle of accelerometer used to estimate the body posture [3].

### 2.4. Statistical Analysis

Multiple regression models for VO_2_ estimation were determined separately for the three accelerometry-based metrics for typical walking and running intensities. Stage-specific VO_2_ was the dependent variable and the respective MAD, MADxyz, or MM values and the test type served as the independent predictor variables. The test type was a dummy variable (track test = 0 and treadmill test = 1). In addition, the receiver operator characteristics (ROC) analysis was used to find the optimal cut-points for moderate, vigorous, and very vigorous PA for MAD, MADxyz, and MM metrics. The cut-point between light and moderate PA was set to 3.0 MET (in terms of oxygen consumption 1 MET = 3.5 mL (O_2_)/kg/min), between moderate and vigorous PA to 6.0 MET, and between vigorous and very vigorous PA to 9.0 MET. The Kolmogorov—Smirnov statistic was used to determine the optimal cut-points.

All statistical analyses were conducted using the statistics software (IBM SPSS Statistics for Windows, Version 27.0, Armonk, NY, USA).

## 3. Results

Table 1 shows the participant characteristics in the track and treadmill test groups. On average, both groups were normal weight, but the treadmill group was 11.6 years younger and 4.7 cm taller than the track group. The age-difference was statistically significant.

Figure 2 shows the stage-specific mean values of accelerometry-based metrics plotted against the locomotion speed and the measured VO_2_. The distinct gap in the MAD, MADxyz, and MM values between speeds from 6 to 8 km/h and VO_2_ from 22 to 30 mL/kg/min coincides with the transition between typical walking and running speeds. In addition, the VO_2_ points displayed different slopes for typical walking and running speeds. Therefore, the regression analysis was conducted separately to points having VO_2_ values less than 25 mL/kg/min and at least 25 mL/kg/min. The measurement points having VO_2_ values less than 25 mL/kg/min showed the range of MAD values from 48 m*g* to 562 m*g*, MADxyz values from 92 m*g* to 684 m*g*, and MM values from 5 m*g* to 138 m*g*. The points having VO_2_ values of at least 25 mL/kg/min showed the range of MAD values from 414 m*g* to 1310 m*g*, MADxyz values from 503 m*g* to 1696 m*g*, and MM values from 73 m*g* to 980 m*g*. The measured VO_2_ values ranged from 7.7 mL/kg/min (2.2 METs) to 68.9 mL/kg/min (19.7 METs).

Table 2 shows the results of the regression analysis separately for typical walking and running intensities VO_2_ < 25 mL/kg/min and VO_2_ ≥ 25 mL/kg/min, respectively. The MAD metric had the best performance for the walking intensities, explaining 86% of the variation in VO_2_ without significantly depending on the test type. For the running intensities, its predictive accuracy was substantially poorer. The MADxyz metric had the best overall performance, explaining 84% and 63% of the variation in VO_2_ during walking and running, respectively. The test type was, however, significantly associated with the predicted VO_2_ during running. The MM metric provided consistent predictive accuracy of VO_2_, explaining 71% and 69% of the variation in VO_2_ during walking and running, respectively, but the test type was significantly associated with the predicted VO_2_ during running.

Figure 3, Figure 4 and Figure 5 show the scatter plots (correlation) and Bland—Altman difference plots between the measured and predicted VO_2_ for MAD, MADxyz, and MM metrics, respectively. In general, the differences were higher for typical running intensities than walking intensities for all metrics. Although the MAD metric had the best and consistent performance for walking intensities, at running intensities, it tended to overestimate the lower VO_2_ values and underestimate the higher VO_2_ values.

Table 3 shows the results of the ROC analysis and optimal cut-points for moderate (3 MET), vigorous (6 MET), and very vigorous (9 MET) PA separately for MAD, MADxyz, and MM metrics. The sensitivity and specificity values were mainly higher than 90%, ranging from 91.6% to 99.4% and from 88.1% to 100%. For each metric, the optimal cut-points for separating different intensity levels from each other differed between the track and treadmill tests and the values depended on the intensity level. In six cases, the track test gave the same cut-point as the pooled dataset. In one case, the treadmill test gave the same cut-point as the pooled dataset, whereas in two cases, the pooled data set had unique cut-points compared to the track and treadmill tests. Of note, the treadmill test data contained 261 datapoints and the track test 137 datapoints, which can give more weight to the treadmill test in the analysis of the pooled dataset.

## 4. Discussion

All three MAD-based accelerometry metrics evaluated in the present study showed characteristic, relatively strong associations with oxygen consumption during locomotion over a wide range of intensities. Mutual to them was that the associations were stronger for typical walking intensities than for running intensities. For the latter, the variation in measured data was wide whereas the test type conferred a significant effect on the prediction of oxygen consumption, except for the conventional MAD metric.

The MAD metric showed the best performance for walking intensities, but the relationship between the accelerometer values and oxygen consumption virtually plateaued during running in the treadmill test. This means that the MAD values did not virtually increase despite the substantially increased intensity. A similar plateau effect during treadmill running was recently observed by Chen et al. [14]. However, in the present study, only one participant showed a negative relationship between the VO_2_ and MAD metric, that is, the MAD values decreased with increasing VO_2_. Individual physical fitness is likely to contribute to this plateau. Those, who had higher MAD values at lower running speeds, were also the first ones to drop out from the test. These individual results likely compromised the performance of the MAD metric during running. The standard error of VO_2_ estimation was over 7 mL/kg/min (corresponding to two METs), indicating a relatively poor accuracy of energy consumption for the MAD metric at high speeds of locomotion. On the other hand, the MAD metric performs accurately when it comes to the classification of PA into intensity categories, for example, into low-intensity PA or moderate-to-vigorous PA (MVPA). These categories, in addition to sedentary time, are commonly used in scientific literature addressing the associations of PA with various health outcomes and PA recommendations [19,20].

The MADxyz metric had good performance for both walking and running intensities. However, the test type confounded the VO_2_ prediction at running speeds, but its impact remained smaller than that with the MM metric. The VO_2_ was 1.8 mL/kg/min lower on average (corresponding to half MET) during treadmill running than during track running at the same MADxyz level. While the MM metric seemed to show the best prediction of VO_2_ for running speeds, the test type conferred a significant impact on the results. VO_2_ during treadmill running was 4.1 mL/kg/min lower (corresponding to more than one MET) than during track running at the same MM level. These test type-related errors can be problematic in free-living assessments of PA intensity because the actual conditions cannot be known a priori or are nearly impossible to measure, especially in studies with a great number of participants.

Treadmill walking qualitatively and quantitatively resembles overground walking. However, when the walking speed is matched, the typical stride length is shorter, and the cadence is higher on the treadmill than on overground [21]. Additionally, overground walking has more variability in temporal rhythm and other gait parameters, such as trunk velocity [22], but VO_2_ is similar during treadmill and overground walking when the speed is the same [23]. In the present track and treadmill tests, the accelerometry-based metrics were at the same level for typical walking speeds. The previously observed differences in the accelerometer outputs between overground and treadmill walking may be attributed to count-based algorithms [5,6]. The count-based metrics used in the previous studies are affected by a band-pass filter, which attenuates the acceleration signal substantially outside the frequency range of 0.25 Hz to 2.5 Hz [24]. Thus, the differences in the preferred gait frequency can modify the accelerometer output and account for the diverging results of the count-based algorithms.

The treadmill and overground running have similar submaximal VO_2_ at lower intensities. Roughly a 1% grade can be used in the treadmill test at running velocities corresponding to at least 80% of the maximal VO_2_ or speeds over 13 km/h to apply more precisely the treadmill-based assessment of running to real-life conditions [25]. However, the biomechanics of running differ between the treadmill and overground tests. On the treadmill, the belt moves the supporting leg under the body, whereas on the overground, the body moves over the supporting leg [26]. Vertical displacement during the entire gait cycle is lower, and the ground contact time is longer during treadmill running [26]. In addition, a person’s inexperience in treadmill running can result in a higher stride frequency and shorter stride length that, in turn, reduce braking and propulsive forces [26]. Humans also optimize their leg stiffness and ground contact to minimize the metabolic cost of running [27]. Apparently, each accelerometer-based metric is differently sensitive to different gait parameters. Therefore, variations in the running conditions can impose specific effects on the accelerometry metrics. In this study, the lower MAD values during the treadmill running may be explained by the lower vertical displacement during the entire gait cycle. The influence of the test type was confirmed by the ROC analysis as clear differences in the cut-points at different intensity levels.

In the present study, the sensor placement was on the side of the hip. The indoor track with banked curves conferred a marginal effect on the accelerometer output between the opposite sides [16]. In general, depending on the habitual activity engaged by the individual, movement-generated accelerations at one site do not necessarily represent those at another site. Therefore, placement of the accelerometer in the middle of the low back could be more optimal for evaluating locomotion because it is closer to the body’s centre of gravity than the hip site [5]. Additionally, for walking and running measurements only, other body locations may provide accurate results as well. For example, wrist-worn devices can predict VO_2_ accurately in different running conditions [28]. However, physical behaviour assessments should provide reliable estimates for all types of physical activities and sedentary behaviour, particularly in population-based studies. Thus, the sensor location does not have to be the best at anything, but be good in everything and feasible to use, as recently recommended by Chastin et al. [2].

According to the present findings, the MAD metric had the best performance for VO_2_ levels up to 25 mL/kg/min, and the MADxyz metric for VO_2_ levels of at least 25 mL/kg/min or higher. In practice, the MAD metric can be used for accurate VO_2_ estimation when the MAD value is less than 500 m*g*. When the MAD value exceeds 500 m*g*, the MADxyz metric is preferable. The MM metric showed slightly better performance for running intensities than the MADxyz metric, but it was also the most sensitive to the test type. Consequently, it Is also more sensitive to unknown variations in free-living conditions, such as surface stiffness, making it susceptible to substantial uncertainty. Nevertheless, because of its consistently strong association with VO_2_ over the wide intensity range of locomotion, the utility of the novel MM metric warrants further evaluation.

One essential difference between the evaluated accelerometry-based metrics is that the MAD metric is not sensitive to sensor rotation whereas the MADxyz and MM metrics are. In other words, when the sensor is rotated in place, the acceleration magnitude corresponds to the level of the Earth’s gravity, but the axis-specific values can vary between −1 *g* to +1 *g*. While the MAD metric reflects the sensor noise, the MADxyz and MM metrics can even indicate high-intensity running. This measurement property may turn useful in predicting VO_2_ during different activities, such as household chores or occupational activities, which are conducted mainly in place but require changes in the body position. This calls for further validation studies as well.

Major strengths of the present study are the direct breath-by-breath measurement of incident VO_2_ during both tests with similar metabolic gas analyzers, well-controlled test conditions, and the use of the same accelerometers for data collection. Further, the data analysis was centralized and based on identical, validated algorithms. On the other hand, the participants were not the same in both tests, and the treadmill group was significantly younger. Other limitations pertain to the unknown performance of metrics during low-intensity locomotion (VO_2_ less than 10 mL/kg/min or less than 3 MET), locomotion at different surfaces, and up- or downhill. Assessment of these conditions calls for further studies.

In conclusion, the MADxyz metric had a similar performance to the MAD metric at walking intensity. However, caution is required with the MADxyz metric because of its high sensitivity to changes in the sensor orientation relative to Earth’s gravity vector. A small change in body posture during the epoch can substantially affect the MADxyz value and possibly result in a fallacious estimation of VO_2_. In all evaluated accelerometry-based metrics, the prediction accuracy of VO_2_ was consistently poorer at high intensities than at low intensities corresponding to walking. Depending on the intensity of locomotion, the choice of proper accelerometer metrics and test type may affect the validity of the prediction of incident VO_2_. However, the accuracy of these metrics is sufficient for the proper classification of physical activity into low, moderate, and vigorous intensity categories that is the most common approach to analyze PA data in large population-based studies.

## Figures and Tables

**Figure 1 sensors-23-05073-f001:**
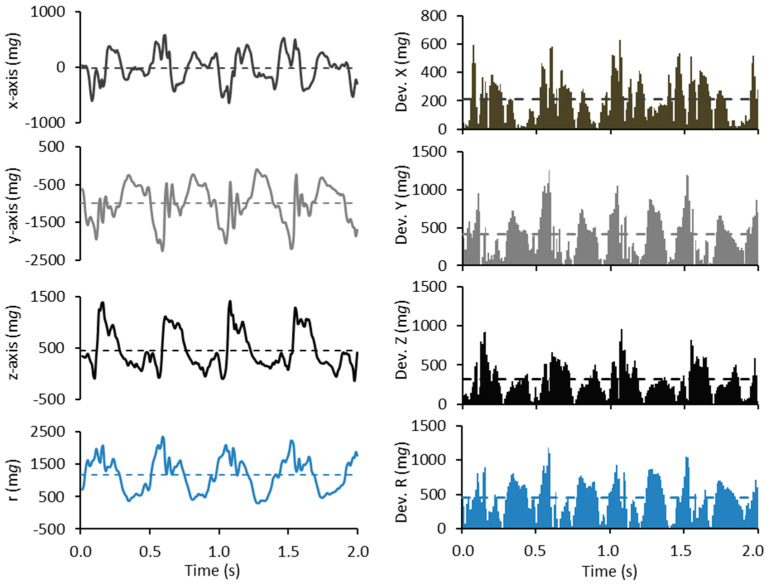
An example of the measured raw acceleration data and processed data during a two-second period of walking. The left panel shows the measured acceleration in x-, y-, and z-directions and their resultant (r) in milligravity units (m*g*). The dashed lines denote the axis-specific and resultant mean acceleration values of the two-second period: −12 m*g* for the *x*-axis, −990 m*g* for the *y*-axis, 453 m*g* for the *z*-axis, and 1169 m*g* for the resultant. The right panel shows incident absolute deviations of axis-specific and resultant accelerations from their sample-wise mean values. The dashed lines denote the respective mean amplitude deviations: 214 m*g* for the *x*-axis, 411 m*g* for the *y*-axis, and 314 m*g* for the *z*-axis, and 456 m*g* for the resultant. The conventional MAD value is 456 m*g*, MADxyz 559 mg (MADxyz = √ (214 m*g*)^2^ + (411 m*g*)^2^ + (314 m*g*)^2^), and MM 80 m*g* (MM = (1169 m*g* – √ (−12 m*g*)^2^ + (−990 m*g*)^2^ + (453 m*g*)^2^).

**Figure 2 sensors-23-05073-f002:**
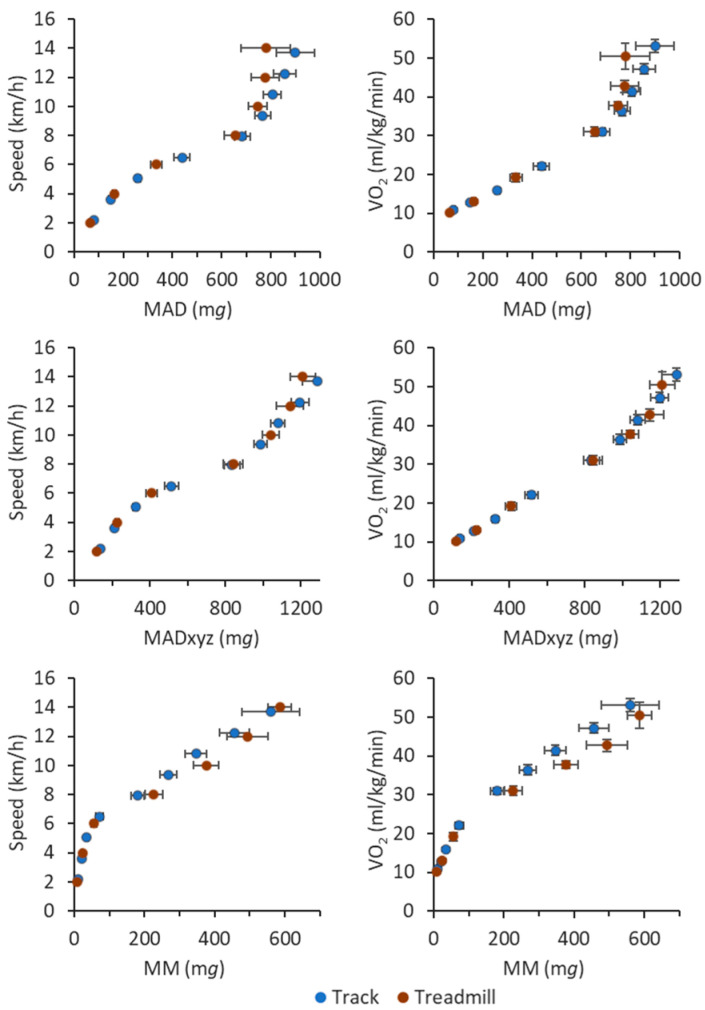
Relationship between the MAD, MADxyz, and MM metrics and the locomotion speed (left panel) and stage-specific VO_2_ (right panel) at different test stages. The blue circles denote the track test and the red circles the treadmill test. The *x*-axis shows MAD, MADxyz, or MM metrics in milligravity (m*g*) units. The error bars denote 95% confidence intervals.

**Figure 3 sensors-23-05073-f003:**
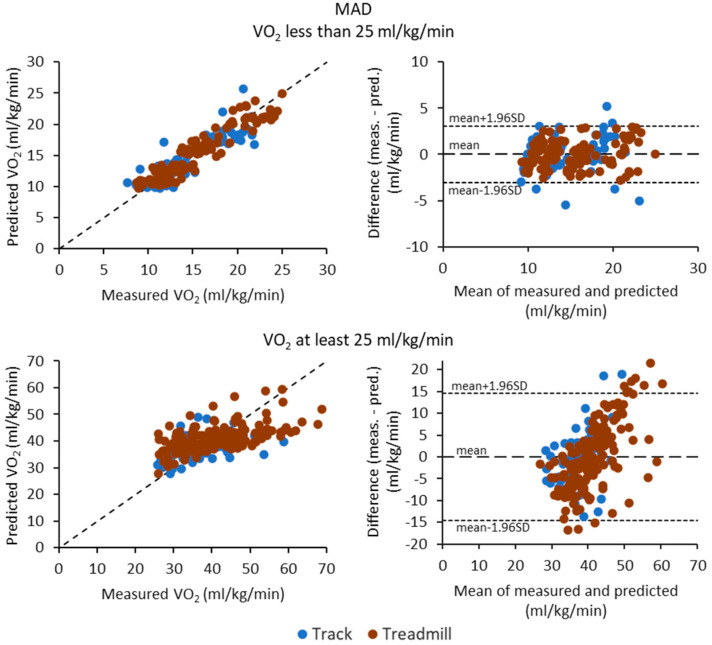
Scatter plots of measured and predicted oxygen consumption (VO_2_) values and Bland—Altman difference plots for the MAD metric. The upper row shows the results for the data points having VO_2_ less than 25 mL/kg/min and lower row for the points having VO_2_ at least 25 mL/kg/min. The dotted lines on the Bland—Altman plot represent the mean difference and the limits of agreement, calculated as the mean difference ± 1.96 times standard deviation (SD) of the differences. The blue circles are for the track test and red circles for the treadmill test.

**Figure 4 sensors-23-05073-f004:**
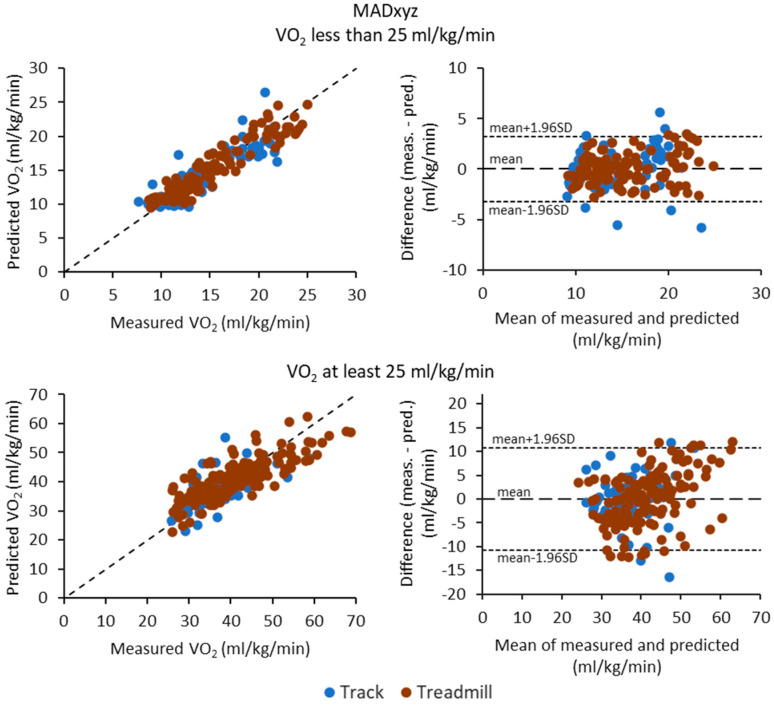
Scatter plots of measured and predicted oxygen consumption (VO_2_) values and Bland—Altman difference plots for the MADxyz metric. The upper row shows the results for the data points having VO_2_ less than 25 mL/kg/min and lower row for the points having VO_2_ at least 25 mL/kg/min. The dotted lines on the Bland—Altman plot represent the mean difference and the limits of agreement, calculated as the mean difference ± 1.96 times standard deviation (SD) of the differences. The blue circles are for the track test and red circles for the treadmill test.

**Figure 5 sensors-23-05073-f005:**
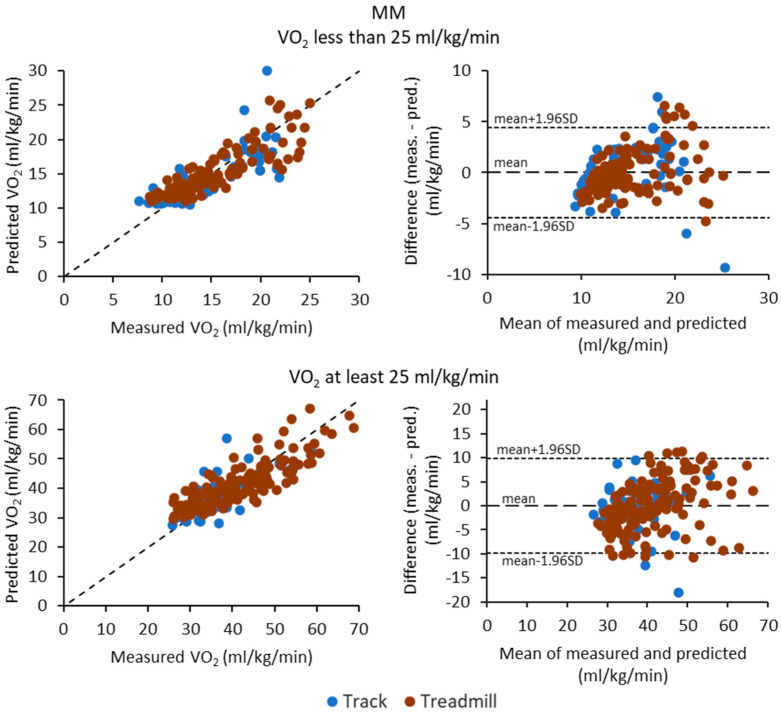
Scatter plots of measured and predicted oxygen consumption (VO_2_) values and Bland—Altman difference plots for the MM metric. The upper row shows the results for the data points having VO_2_ less than 25 mL/kg/min and lower row for the points having VO_2_ at least 25 mL/kg/min. The dotted lines on the Bland—Altman plot represent the mean difference and the limits of agreement, calculated as the mean difference ± 1.96 times standard deviation (SD) of the differences. The blue circles are for the track test and red circles for the treadmill test.

**Table 1 sensors-23-05073-t001:** Participant characteristics (mean, SD).

	Track Test Group	Treadmill Test Group
N	29	24
Women (%)	14 (48.3%)	12 (50.0%)
Men (%)	15 (51.7%)	12 (50.0%)
Age (years)	35.2 (10.8)	23.6 (3.6) *
Weight (kg)	70.4 (12.5)	70.3 (12.0)
Height (cm)	172.1 (9.8)	176.8 (11.6)
BMI (kg/m^2^)	23.4 (2.3)	22.4 (2.2)

BMI is body mass index. SD is standard deviation. * Between-group difference *p* < 0.0001.

**Table 2 sensors-23-05073-t002:** Regression models of oxygen consumption (VO_2_) for walking and running intensities.

	Walking VO_2_ < 25 mL/kg/min		Running VO_2_ ≥ 25 mL/kg/min	
	Coefficient	*p*	Coefficient	*p*
MAD metric				
Constant	8.236	<0.001	13.006	<0.001
Test type	0.010	0.967	−1.168	0.314

MAD	0.031	<0.001	0.035	<0.001

R^2^	86%		32%	
SEE	1.59		7.46	
MADxyz metric				
Constant	6.874	<0.001	5.814	0.005
Test type	−0.051	0.841	−1.805	0.032
MADxyz	0.029	<0.001	0.033	<0.001

R^2^	84%		63%	
SEE	1.68		5.53	
MM metric				
Constant	10.355	<0.001	26.559	<0.001
Test type	−0.563	0.102	−4.138	<0.001
MM	0.146	<0.001	0.041	<0.001

R^2^	71%		69%	
SEE	2.30		5.07	

R^2^ is the coefficient of determination. SEE is the standard error of the estimate. The test type is 0 for the track test and 1 for treadmill test.

**Table 3 sensors-23-05073-t003:** Moderate (3 MET), vigorous (6 MET), and very vigorous (9 MET) physical activity cut-points based on the receiver operator characteristics analysis for MAD, MADxyz, and MM metrics determined from the treadmill test, track test, and pooled data.

		Cut-Point (m*g*)	AUC (95% ci)		Sensitivity	Specificity
MAD	3 MET					
	Treadmill	148.2	0.956	(0.922–0.990)	91.6%	100.0%
	Track	93.6	0.968	(0.947–0.989)	94.0%	100.0%
	Pooled	93.6	0.967	(0.951–0.984)	94.0%	96.7%
	6 MET					
	Treadmill	348.6	0.993	(0.981–1.002)	97.1%	97.1%
	Track	396.3	0.996	(0.992–1.000)	98.7%	96.1%
	Pooled	396.3	0.995	(0.992–0.999)	97.8%	96.5%
	9 MET					
	Treadmill	565.6	0.982	(0.960–1.004)	98.2%	92.7%
	Track	639.4	0.967	(0.947–0.988)	96.6%	89.7%
	Pooled	552.5	0.970	(0.953–0.986)	99.4%	88.1%
MADxyz	3 MET					
	Treadmill	213.6	0.958	(0.926–0.991)	89.9%	100.0%
	Track	146.4	0.979	(0.962–0.995)	96.0%	100.0%
	Pooled	149.2	0.972	(0.957–0.988)	94.6%	96.7%
	6 MET					
	Treadmill	556.2	0.991	(0.981–1.000)	94.2%	98.5%
	Track	466.7	0.996	(0.991–1.000)	97.5%	96.1%
	Pooled	466.7	0.995	(0.991–0.999)	96.9%	96.5%
	9 MET					
	Treadmill	761.2	0.988	(0.971–1.003)	96.4%	95.1%
	Track	846.9	0.981	(0.968–0.995)	94.0%	93.1%
	Pooled	761.2	0.983	(0.973–0.993)	97.7%	90.7%
MM	3 MET					
	Treadmill	12.8	0.961	(0.930–0.992)	94.1%	94.4%
	Track	13.3	0.972	(0.952–0.985)	93.2%	100.0%
	Pooled	13.3	0.969	(0.952–0.985)	93.5%	96.7%
	6 MET					
	Treadmill	111.3	0.990	(0.977–1.002)	94.2%	98.5%
	Track	75.6	0.992	(0.985–0.998)	93.7%	99.0%
	Pooled	75.6	0.990	(0.984–0.996)	93.8%	98.2%
	9 MET					
	Treadmill	216.4	0.989	(0.978–1.000)	92.7%	96.3%
	Track	187.4	0.981	(0.969–0.994)	94.0%	94.5%
	Pooled	187.4	0.984	(0.975–0.993)	94.2%	93.8%

MET = metabolic equivalent (1 MET = 3.5 mL (O_2_)/kg/min; AUC = area under curve; m*g* = milligravity.

## Data Availability

Non-identifiable data are available for research purposes from the corresponding author upon reasonable request.

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
