# Peer review of "Performance of Different Accelerometry-Based Metrics to Estimate Oxygen Consumption during Track and Treadmill Locomotion over a Wide Intensity Range"

_sensors, 2023, doi:10.3390/s23115073_

Round 1

Reviewer 1 Report

The present work shows an interesting proposal to adjust the mathematical equation that can best correlate with the energy consumption produced in an exercise.

In addition, this adjustment is compared with different progressive load tests, performed on a treadmill or on a 200m running track.

All study subjects wear the same accelerometer and a gas analyser that allows the measurement of VO2, breath-by-breath, during the progressive increase test.

In the paper they analyse three mathematical equations (MAD, MADxyz and MM) considering that the MAD formula is most accurate when VO2 is less than 25ml/kg/min and the MADxyz the most accurate when VO2 is greater than 25ml/kg/min.

In the analysis they present the cut-off point values using ROC curves and extract the regression of the VO2 models on each of the formulae at walking and running intensities.

In lines 334-343 of the discussion, reference is made to the difference between treadmill and track running values based on the biomechanical difference in running, longer support time on the treadmill and less vertical displacement when running on the treadmill. This assertion is not supported by biomechanical data and should be studied for validation.

One difference between running on the track and running on the treadmill is in the location of the accelerometer. In this study, subjects wore the accelerometer on the hip, usually on the right, while track runners wore the accelerometer half on the right and half on the left.

When running on the track, the subject moves forward on the horizontal axis, there is a forward translation of the body as a whole, whereas, on the treadmill, the subject remains in the same place, it is the treadmill that moves, so the subject remains in the same place, there is no forward translation of the body as a whole. This fact is not perceptible when the speed of displacement is low, such as walking, but as the speed of displacement increases this factor is more decisive, since the influence of the value of the horizontal axis is more relevant and makes the values of the different mg results different between the measurement on the track and on a treadmill.

In order to be able to compare more accurately the treadmill and the athletics track, the device should be placed in a place that maintains the same biomechanical running conditions, for example, the wrist or the ankle, as its biomechanical performance is not conditioned by the subject's movement but by his or her technical gesture.

A work that has related VO2 and accelerometry values in runners is (Hernando et al., 2018) which they then used in marathon runners, so the biomechanical conditions of the technical gesture are reproduced in the different running conditions. It would be advisable to include this comment in the section on limitations or actions to be considered in a possible new study.

Finally, in general, it is advisable to change VO2 which appears in the text the radical should be presented in subscript, for VO2

Hernando, Carlos, Hernando, Carla, Collado, E.J., Panizo, N., Martinez-Navarro, I., Hernando, B., 2018. Establishing cut-points for physical activity classification using triaxial accelerometer in middle-aged recreational marathoners. PLOS ONE 13, e0202815. https://doi.org/10.1371/journal.pone.0202815

Author Response

Attached please find the revised version of our manuscript. We found the reviewers’ comments and suggestions useful and took them well in our revision. The point-by-point responses to your comments in italics are given below. In the text, the major changes are indicated by red color.

Reviewer 1:

The present work shows an interesting proposal to adjust the mathematical equation that can best correlate with the energy consumption produced in an exercise. In addition, this adjustment is compared with different progressive load tests, performed on a treadmill or on a 200m running track. All study subjects wear the same accelerometer and a gas analyser that allows the measurement of VO2, breath-by-breath, during the progressive increase test. In the paper they analyse three mathematical equations (MAD, MADxyz and MM) considering that the MAD formula is most accurate when VO2 is less than 25ml/kg/min and the MADxyz the most accurate when VO2 is greater than 25ml/kg/min. In the analysis they present the cut-off point values using ROC curves and extract the regression of the VO2 models on each of the formulae at walking and running intensities.

We greatly appreciate your overall summary of our paper. Thank you for your relevant comments. Please find our response in italic below your comments.

In lines 334-343 of the discussion, reference is made to the difference between treadmill and track running values based on the biomechanical difference in running, longer support time on the treadmill and less vertical displacement when running on the treadmill. This assertion is not supported by biomechanical data and should be studied for validation.

One difference between running on the track and running on the treadmill is in the location of the accelerometer. In this study, subjects wore the accelerometer on the hip, usually on the right, while track runners wore the accelerometer half on the right and half on the left.

When running on the track, the subject moves forward on the horizontal axis, there is a forward translation of the body as a whole, whereas, on the treadmill, the subject remains in the same place, it is the treadmill that moves, so the subject remains in the same place, there is no forward translation of the body as a whole. This fact is not perceptible when the speed of displacement is low, such as walking, but as the speed of displacement increases this factor is more decisive, since the influence of the value of the horizontal axis is more relevant and makes the values of the different mg results different between the measurement on the track and on a treadmill.

In order to be able to compare more accurately the treadmill and the athletics track, the device should be placed in a place that maintains the same biomechanical running conditions, for example, the wrist or the ankle, as its biomechanical performance is not conditioned by the subject's movement but by his or her technical gesture.

A work that has related VO2 and accelerometry values in runners is (Hernando et al., 2018) which they then used in marathon runners, so the biomechanical conditions of the technical gesture are reproduced in the different running conditions. It would be advisable to include this comment in the section on limitations or actions to be considered in a possible new study.

Quite right. As you stated, the main difference between the treadmill and track running pertains to the fact that on the treadmill the belt is moving whereas on the track the walker/runner is moving. We have addressed this issue in the discussion (page 13, lines 324-326)

 We have also added two new paragraphs about the location of the accelerometer and measurement properties (page 13, paragraphs 2 and 4). It is true that the side of the hip is not an optimal location for an accelerometer, because it cannot detect differences between the legs. However, in long-term continuous monitoring of physical activity behavior the side of the hip is less obstructive than the middle of the back. It is also true that the wrist-worn devices have usually the best user satisfaction and they have high accuracy to estimate VO2 and we added the reference to the study by Hernandez et al. you mentioned. However, wrist-worn devices cannot separate lying, sitting, and standing from each other, which are the most typical activities in population-based studies which need to be assessed as well besides physical activity.

Finally, in general, it is advisable to change VO2 which appears in the text the radical should be presented in subscript, for VO2

Quite agree. Corrected as advised throughout the text.

Reviewer 2 Report

I enjoyed seeing this comparison of different measurement and data analyis methods, which all seem to have been carefully performed. I have only a few relatively minor suggestions.

The notation mg can easily be confused with milligrams. Possibly it could be solved by making "g" in italic (since it is a physical property rather than an ordinary unit). Alternatively all results could be presented in terms of "g" since the graphs are using scales with 100's of milli-g's.

Obviously a combination of the xyz accelerometer sensors give information about the angle of the sensors, while using the absolute value loses some of the sensitivity. I suggest the authors at least give some hints about the possibility to include rotation sensor data, as well,

I am confused about the note in the "Author contribution", claiming that two authors are credited with "funding acquisition", while the next section – "Funding" – states that no extra funding was received.

Author Response

Attached please find the revised version of our manuscript. We found the reviewers’ comments and suggestions useful and took them well in our revision. The point-by-point responses in italics to your comments are given below. In the text, the major changes are indicated in red color.

Reviewer 2:
I enjoyed seeing this comparison of different measurement and data analyis methods, which all seem to have been carefully performed. I have only a few relatively minor suggestions.

We greatly appreciate your overall opinion of our paper. Thank you for your comments.

The notation mg can easily be confused with milligrams. Possibly it could be solved by making "g" in italic (since it is a physical property rather than an ordinary unit). Alternatively all results could be presented in terms of "g" since the graphs are using scales with 100's of milli-g's.

We agree that the notation ‘mg’ can be confusing. The ‘g’ is now changed to italic throughout the text.

Obviously a combination of the xyz accelerometer sensors give information about the angle of the sensors, while using the absolute value loses some of the sensitivity. I suggest the authors at least give some hints about the possibility to include rotation sensor data, as well,

We have also added a new paragraph (page 13, paragraph 4)  which discusses  the influence of rotations on metrics. It is likely that the low MAD value and high MADxyz and MM values can be useful. At least, they can it tell that the body position is changing but not moving.  

I am confused about the note in the "Author contribution", claiming that two authors are credited with "funding acquisition", while the next section – "Funding" – states that no extra funding was received.

Thank you for bringing up this inconsistency. The ‘Author contribution' and ‘Funding’ statements are now clarified as follows. We removed Funding acquisition from Author contributions. The funding of this project was based on the normal funding of both organizations without external funding. We hope that this removes the confusion.